# Social Support Mediates the Association between Health Anxiety and Quality of Life: Findings from a Cross-Sectional Study

**DOI:** 10.3390/ijerph191912962

**Published:** 2022-10-10

**Authors:** Marta Ciułkowicz, Błażej Misiak, Dorota Szcześniak, Jolanta Grzebieluch, Julian Maciaszek, Joanna Rymaszewska

**Affiliations:** 1Department of Psychiatry, Wroclaw Medical University, 50-367 Wroclaw, Poland; 2Department of Population Health, Wroclaw Medical University, 50-367 Wroclaw, Poland

**Keywords:** cyberchondria, health anxiety, quality of life, perceived social support

## Abstract

This study aimed to test if perceived social support and cyberchondria mediate the association between health anxiety and quality of life (QoL) in a nonclinical sample. Cross-sectional research involved adult internet users (*n* = 538) between 16 May 2020 and 29 December 2020 in Poland who completed self-report questionnaires, including the cyberchondria severity scale (CSS-PL), the short health anxiety inventory (SHAI), the multidimensional scale of perceived social support (MSPSS) and the quality of life scale (QOLS). A mediation analysis was performed to examine the direct effects of health anxiety on cyberchondria, perceived social support and quality of life. Likewise, the effects of cyberchondria and perceived social support on QoL were analyzed. Hence, indirect effects of health anxiety on QoL through cyberchondria and perceived social support were explored. Health anxiety significantly impaired QoL both directly and indirectly through low-perceived social support. Perceived social support partly mediated the association between health anxiety and QoL. Cyberchondria did not have a significant direct effect on the latter. Thus, cyberchondria did not mediate the relationship between health anxiety and QoL. Boosting-perceived social support may mitigate the detrimental effect of health anxiety on QoL. Cyberchondria was not found to have a significant effect on QoL in contrast to health anxiety alone.

## 1. Introduction

Quality of life (QoL) is a meaningful and subjective measure, encompassing the interplay between expectations and the actual experience of existence [1]. Despite health being recognized as a principal contributor, the overall concept of QoL transcends the simple estimation of physical and mental states. It includes financial, interpersonal, societal, recreational and fulfilment aspects of living [2] against a given cultural background and value system [3]. Proper QoL facilitates reaching full individual potential as well as being able to be translated to tangible epidemiological results, such as a lower mortality risk [4]. Therefore, promoting good QoL, irrespective of the life stage, has been recognized as a challenge by major health organizations globally [5,6,7].

One of the symptoms that could markedly compromise individual QoL during an epidemiological crisis is health anxiety [8]. This term refers to a continual, unfounded and excessive fear of a possible medical condition. It may overlap with or underlie a wide range of anxiety disorders, such as, for instance, obsessive–compulsive disorder or somatic symptom disorder [9]. Preoccupation with health-related worry or misperception of bodily symptoms may trigger a number of safety behaviors, including the excessive use or avoidance of medical services [10]. The persistence of these behaviors reinforces them and serves as a risk of severe impairments in daily functioning.

Notably, one of the factors decreasing anxiety levels may be perceived social support [11]. The term social support refers to the health-promoting dimensions of human interactions. Received social support draws upon the objective quantity and quality of the social support provided. On the other hand, perceived social support is conceptualized as a self-evaluation of provided support and individual satisfaction with such support [12]. Being mindful of key differences between those terms is crucial, as the correlation between received and perceived social support is not necessarily obvious [13,14]. Notably, perceived social support is a more stable yet modifiable measure, anchored in one’s personality structure [15,16], and was found to predict mental health to a greater extent than objective social support [13]. The pivotal role of interpersonal relations up against an epidemiological crisis was mirrored in the careful use of pandemic-related nomenclature. For example, it has been advised to replace “social isolation” and “social distancing” with “physical distancing” not to discourage connectedness [17,18,19]. Data regarding the directionality of prediction between perceived social support and anxiety remains relatively scarce. One scenario is that low-perceived social support may increase the level of anxiety [20]. However, from the perspective of the cognitive–behavioral theory [21], mutual interactions can appear between both constructs as they relate to certain cognitive schemas that result in particular outcomes, such as, for instance, anxiety. Adding the continuous nature of perceived social support, the effect of health anxiety on perceived social support could be anticipated and addressed. Moreover, perceived social support was observed to be an independent predictor of quality of life [22].

Staying in touch without making actual contact would not be possible if not for the internet. Apart from the apparent advantages of online interactions [23], such as buffering anxiety among the isolated [24] and the lonely [25], the negative backlash of the massive pandemic-related information exchange is not to be overlooked. Vismara et al. [26] analyzed a sample of 572 internet users and concluded that the internet was the most popular source to look for medical information amid the COVID-19 pandemic. What is more, the surge in such searches was found in almost one-third of the respondents since the pandemic outbreak. Some internet users may start searching for valid medical sources and gradually distance themselves from evidence-based data [27]. Amateur and emotion-saturated health information was observed to have varied and adverse effects on mental health [28,29] in predisposed individuals. Anxiety-triggered, persistent and unrestrained searching for medical information on the internet connected with lasting or even increased distress is described as cyberchondria [27,30]. Cyberchondria was found to be a behavioral pattern conceptually relevant to health anxiety, hypochondriasis, general anxiety, anxiety about COVID-19, metacognitive beliefs about anxiety, obsessive–compulsive disorder and problematic usage of the internet [26,31]. However, the overlap of cyberchondria and health anxiety deserves special emphasis. Even though the correlation between health anxiety and cyberchondria can be supported by the meta-analysis performed by McMullan et al. [32], the temporal precedence and directionality of their relationship are debatable. Menon et al. [33] suggests that health anxiety is not a sine qua non factor for cyberchondria, as well as it not being confirmed that cyberchondria is a risk factor or a maintaining factor for health anxiety. In this light, cyberchondria and health anxiety should be treated as separate phenomena [33], and their relationship may be potentially bilateral. Nevertheless, minding the fact that the recent study by Nadeem et al. [34] shows that health anxiety is a positive predictor of cyberchondria and anxious individuals are more prone to search the internet for medical data [35,36], the examination of health anxiety was prioritized as a predictor variable needing to be highlighted.

Moreover, the importance of age, gender and education with reference to the aforementioned constructs is supported by the state-of-the-art. As the styles of internet use may be age-dependent [37], the older population could be less prone to cyberchondria [38,39]. On the other hand, the onset of severe health anxiety is more prevalent in senior populations [40]. The impact of perceived social support was observed to vary between different age groups [41]. Additionally, internet use and health anxiety patterns may also be related to gender and education [42,43]. Atkinson et al. [44] propose that educated females are more willing to search for medical data on the internet compared to males and those without bachelor’s degrees. Moreover, some studies imply higher cyberchondria severity within females [38,45].

Associations between cyberchondria and quality of life are novel objects of interest [46,47,48,49] and seek further examination. Additionally, although there is evidence that a high level of health anxiety is associated with low QoL, the exact mechanisms underlying this association remain unknown. A better understanding of this matter might help to develop interventions aiming to improve the general and psychological well-being of individuals experiencing health anxiety. Therefore, in this study, we investigated the mechanisms linking health anxiety with QoL. Specifically, we tested the hypothesis that the association between health anxiety and QoL is mediated by cyberchondria and perceived social support (Figure 1).

## 2. Materials and Methods

### 2.1. Participants

The cross-sectional online survey was distributed between 16 May 2020 and 29 December 2020 in Poland. The snowball sampling method was applied to involve an adult sample of the Polish population of internet users. This technique enabled the prompt collection of sensitive and confidential data regardless of pandemic-related restrictions. The questionnaires were collected in line with the computer-assisted web interview (CAWI) method [50]. They were spread via social media, e-mail addresses and, to a lesser extent, using social backgrounds of the researchers. Participants were informed about its confidential and anonymous character. This information was stated at the very beginning of the questionnaire. Participants confirmed reaching adulthood as well as being familiar with the study’s description, goal and terms by submitting a filled survey. Only completed questionnaires were analyzed. The Ethics Committee at Wroclaw Medical University (Poland) approved the study protocol (approval number: 286/2020). The study was performed in agreement with the principles of the Declaration of Helsinki. The paper’s structure is consistent with STROBE statements for reporting cross-sectional studies [51].

### 2.2. Measures

#### 2.2.1. The Quality of Life Scale (QOLS)

The Polish version of the QOLS, developed by Burckhardt [52,53] is a 16-item, self-administered questionnaire. It transcends health-related quality of life as it explores 5 conceptual categories: material and physical well-being (items 1 and 2), relationships with other people (items 3, 4, 5 and 6); social, community and civic activities (items 7 and 8); personal development and fulfillment (items 9, 10, 11 and 12) and recreation (items 13, 14, 15 and 16). Possible answers range on a 7-point scale from delighted (7) to pleased (6), mostly satisfied (5), mixed (4), mostly dissatisfied (3), unhappy (2) and terrible (1). The total score ranges from 16 to 112, with higher scores indicating better QoL. The total score was found to correlate with both the physical health status as well as disease measures [52]. Cronbach’s alpha was 0.93 in the present study.

#### 2.2.2. The Short Health Anxiety Inventory (SHAI)

The Polish adaptation of the SHAI [54,55] is a 16-item, self-reported questionnaire recording two categories of hypochondriasis: illness likelihood (IL) and negative consequences of an illness (NC). However, the total score can also be considered. Each item is based on four statements related to the preceding 6 months: no symptoms (0), mild symptoms (1), severe symptoms (2) and very severe symptom (3) of clinical hypochondriasis. An optimal cut-off score was established at 20 points and could be characterized by a sensitivity of 79.3% and specificity of 78.0% in differentiating hypochondriasis from other anxiety disorders. Cronbach’s alpha of the Polish adaptation was described as excellent, as it exceeded 0.90 [54]. In the current research, it was estimated at 0.92.

#### 2.2.3. Cyberchondria Severity Scale (CSS)

The CSS is a 33-item self-report that enables the complex assessment of cyberchondria [39,56]. Items are grouped in 5 subscales that include: compulsion (items 3, 6, 8, 12, 14, 17, 24 and 25), distress (items 5, 7, 10, 20, 22, 23, 29 and 31), excessiveness (item 1, 2, 11, 13, 18, 19, 21 and 30), reassurance (items 4, 15, 16, 26, 27 and 32) and mistrust of medical professionals (items 9, 28 and 33). The answers are based on a 5-point Likert scale (1—never; 2—rarely; 3—sometimes; 4—often; and 5—always). Cronbach’s alpha for the Polish adaptation was consistent with the original version [56] and ranged between 0.75 and 0.95. It was evaluated to be 0.90 in this study.

#### 2.2.4. The Multidimensional Scale of Perceived Social Support (MSPSS)

The MSPSS is a 12-item measure of perceived social support perceived from one’s inner circle [57,58]. The items are grouped into three subscales regarding the source of the support: family, friends and significant others. Answers are based on a 7-point Likert scale ranging from strongly disagree (1) to strongly agree (7). Total and subscale scores can be calculated. Greater scores indicate higher perceived social support. Cronbach’s alpha for the Polish adaptation was established at 0.893 [58]. In our research, it was calculated to be 0.94.

### 2.3. Statistics

Associations between continuous variables were tested with two-tailed Pearson correlation. The Kolmogorov–Smirnov test was used to analyze the normality of data distribution. Results of bivariate tests were considered significant if the *p*-value was <0.05. The PROCESS macro [59] was used to test parallel mediation models (Figure 1). Health anxiety was included as an independent variable and quality of life was an outcome variable. Cyberchondria and perceived social support were included as mediators. The mediation analysis aimed to explore the direct effects of health anxiety on cyberchondria (a1), perceived social support (a2) and quality of life (c). Moreover, such effects were investigated regarding cyberchondria and quality of life (b1), as well as perceived social support and quality of life (b2). Thereafter, the indirect effects of health anxiety on quality of life through cyberchondria (a1b1) and perceived social support (a2b2) were investigated. Age, gender and education were added as covariates. Direct and indirect effects were examined using bootstrap calculation with 5000 samples. Mediation was considered significant if the 95% CI of an indirect effect did not include a zero. Before performing mediation analyses, assumptions of the linear regression analysis were tested. These included: (1) linearity of the relationship between independent variables and dependent variables; (2) normal distribution of the residuals; (3) homoscedasticity of the residuals; (4) uncorrelatedness of the residuals; (5) absence of multicollinearity; (6) a lack of extreme outliers. Linearity was assessed by inspecting the partial scatterplots. A normal distribution of the residuals was checked with visualizing histograms and P–P plots. Homoscedasticity was analyzed by plotting the regression studentized residuals and the regression standardized predicted values. The Durbin–Watson statistics were tested to analyze the uncorrelatedness of residuals. Values between 1.5 and 2.5 were interpreted as indicating no first-order autocorrelation [60]. Multicollinearity was evaluated using the variance inflation factor (VIF). VIF values > 4 were considered as showing significant multicollinearity [61]. Finally, case-wise diagnostics were carried out and standardized residuals located outside 3 standard deviations were interpreted as outliers. All analyses were performed with Statistical Package for Social Sciences, version 20 (SPSS Inc., Chicago, IL, USA) [62].

## 3. Results

The descriptive statistics of the study sample are presented in Table 1. Data from 538 surveyed individuals were analyzed. The mean age of the respondents was 36.64 years (SD = 12.55; range = 18–75). There was a predominance of females and subjects with higher education.

The bivariate correlations are presented in Table 2. Health anxiety was significantly and positively correlated with cyberchondria. Moreover, there were significant negative correlations between health anxiety and perceived social support and QoL. In turn, QoL was significantly and positively correlated with perceived social support, as well as being negatively correlated with health anxiety and cyberchondria. In addition, older age was associated with greater QoL. A gender-sensitive analysis showed no significant correlation between perceived social support and health anxiety, or between perceived social support and cyberchondria among males. This contrasted with the significant correlation between these two constructs in a female group, as well as in the whole sample.

The bivariate correlations regarding education level can be found in Table 3. QoL significantly correlated with age, and cyberchondria did not correlate with perceived social support in the higher educated, unlike other education groups.

The results of the mediation analysis are shown in Table 4. All assumptions of the multiple regression analysis were met. There were significant direct effects of health anxiety on cyberchondria and perceived social support. Similarly, the direct effect of perceived social support on QoL was also found to be significant. However, cyberchondria was not directly associated with QoL. There was a significant indirect effect of health anxiety (through social support) on QoL. In turn, the indirect effect of health anxiety on QoL (through cyberchondria) appeared to be insignificant.

## 4. Discussion

The main ambition of this cross-sectional study was to explore the relationships between QoL, health anxiety, cyberchondria and perceived social support. Particular attention was paid to if cyberchondria and perceived social support mediated associations between health anxiety and QoL. Findings from the present study implied that health anxiety negatively affects QoL both directly and indirectly through poor perceived social support. In other words, perceived social support partially mediates the association between health anxiety and QoL. Although cyberchondria was not found to serve as a significant mediator between health anxiety and QoL, it appeared to be associated with health anxiety.

Our study explored cyberchondria and health anxiety as separate but strongly correlated phenomena. It aligned with existing evidence suggesting that health anxiety is essential for the development of severe cyberchondria [63]. Having stated that, health anxiety and cyberchondria may have similar but not identical properties. Despite treating cyberchondria as a distinct construct of health anxiety, it still sparks debate, with current research supporting various effects of those concepts on QoL. The literature consistently confirms that health anxiety [64], hypochondriasis [65], obsessive–compulsive disorder [66], generalized anxiety, panic disorders [67] or internet addiction [68] impair life quality, although this was not evident in the case of cyberchondria in the current study. This observation stands in line with the study by Mathes et al. [46], who concluded that individuals experiencing cyberchondria did not experience impaired quality of life. A recent study by Ambrosini et al. [49] somehow complemented this observation by indicating that the impact of cyberchondria on QoL might be fully mediated by obsessive–compulsive manifestations, as well as Internet addiction. These mediators were also described as significant when relations between cyberchondria and health anxiety were noted. By this rationale, it could be hypothesized that the element of compulsive internet searching for medical information could serve as an unadaptive coping mechanism fueled by reassurance seeking [27].

What is more, the model based on the cognitive–behavioral theory suggests that such transient feelings of relief right after receiving reassurance may, paradoxically, assist in sustaining such maladaptive behavior underpinned by health anxiety [69]. Therefore, the compulsive features of cyberchondria may be crucial in determining the impact of cyberchondria on QoL. Of note, the actual consequences of searching the internet for solace remain unpredictable, as 40% of individuals may encounter escalated anxiety [70]. From a different angle, Vismara et al. [26] observed that cyberchondria negatively correlated with QoL. The authors found that individuals with baseline poor QoL could be more prone to develop cyberchondria, or that cyberchondria may impair QoL.

At the same time, Rahme et al. [48] found social support to be a substantial factor in mitigating the mentioned factors’ detrimental effects on QoL. This remark corresponded, to the same extent, with our results, showing that perceived social support mediated the association of health anxiety with QoL. Such a relation between health anxiety and perceived social support provided a novel finding, as well as complemented the existing literature on the anxiety spectrum and perceived social support. It can be analyzed in two ways, regarding its physiological and psychological background. Enhanced support against a stressor may stem from the oxytocin-moderated biobehavioral reaction that facilitates mutual protection and creates novel social networks to reinforce survival [71,72].

To put it differently, in the face of a threat, people tend to affiliate with others to minimize stress, anxiety and affective responses that can potentially impair QoL. From the psychological point of view, social support supplies a person with resources critical to cope with a crisis [73] and facilitates resilience [74]. It covers delivering emotional, informational, material, instrumental and spiritual help [75]. Moreover, in the current study, significant gender-related differences were found when correlations between perceived social support and cyberchondria, as well as perceived social support and health anxiety, were concerned. Both cyberchondria and health anxiety were not found to correlate significantly with perceived social support in the male group in contrast with females. It could be suspected that such divergences relate to varied socialization and expectations regarding roles among men and women [76].

Certain limitations characterized the current study. The cross-sectional design did not allow us to draw casual conclusions. That stated, the study should be replicated using longitudinal or experimental designs. Self-assessments may have posed a risk of both information bias. Likewise, snowball sampling may have limited the study’s validity due to a lack of a random selection of participants.

Additionally, even though online data sampling used in the present study had the advantage of collecting large datasets [50], the exact number of individuals approached for participation was not recorded. Thus, the response rates remain unknown. It is also important to note that most responses were received from women and people with higher education. The multidimensional and complex nature of the explored phenomena could potentially be related to a residual confounding bias. For instance, high baseline health anxiety is not a prerequisite for seeking medical data online [77,78,79]. Hence, the study could have been extended to explore trait anxiety. It seems reasonable to incorporate this variable into future research, as perceived social support corresponds to personality traits. There are indications that this approach could be well founded also regarding cyberchondria [80,81]. The current research was triggered by the massive and rapid surge in Internet use as a reaction to the global health crisis.

Nonetheless, the explored phenomena might have been sensitive to pandemic-related circumstances such as the limited provision of daily services or altered social interaction patterns. Minding the fact that the data were collected within a few months, it could be speculated that the experiences of the predisposed individuals were not linear during the study period. Having stated that, the results should be interpreted and generalized with caution. As a final point, even though the current study was performed during the COVID-19 pandemic, it was not based on any objective pandemic-related measures. Moreover, the administered questionnaires, apart from SHAI, did not specify a particular period during which the individuals experienced the measured constructs. The current research did not foster any conclusions regarding relations between health anxiety, perceived social support, cyberchondria, quality of life and the current epidemiological crisis. However, exploring such associations might be a promising research direction.

## 5. Conclusions

In conclusion, the main findings of this study implied that health anxiety can contribute to low QoL directly and indirectly through a lower use of social support resources. Enhancing the individual perception of having sufficient material and psychological support may mitigate the detrimental effect of health anxiety on QoL. These findings could be used to inform public policies that aim to create support strategies for individuals with high levels of health anxiety. However, additional studies, especially those adopting longitudinal designs based on representative populations and exploring a wide range of psychopathology and disrupted behavioral patterns, are still needed to confirm causal associations.

## Figures and Tables

**Figure 1 ijerph-19-12962-f001:**
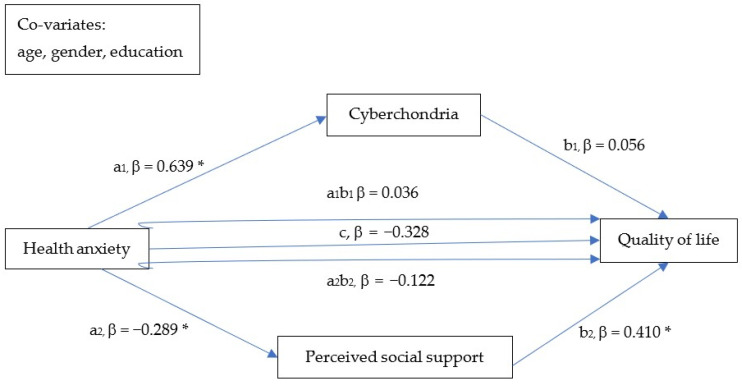
The multiple mediation model tested in the study. Significant effects (95% CI did not include zero) were marked with asterisks.

**Table 1 ijerph-19-12962-t001:** General characteristics of the sample (*n* = 538).

Variable	*n* (%)	Mean ± SD	Median (Range)	Skewness	Kurtosis	Kolmogorov–Smirnov Test
Age, years		36.65 ± 12.55	34.0 (59.0)	0.768	−0.006	*p* < 0.001
Gender, males	100 (18.6)					
Education, higher	422 (78.4)					
CSS-PL total score		60.90 ± 17.58	57.5 (99.0)	0.817	0.643	*p* = 0.191
SHAI total score		15.53 ± 8.80	14.0 (51.0)	1.104	1.693	*p* = 0.103
MSPSS total score		70.23 ± 15.75	76.0 (72.0)	−1.408	1.709	*p* = 0.058
QOLS total score		79.67 ± 15.13	80.5 (91.0)	−0.660	0.991	*p* = 0.088

Abbreviations: SHAI—the short health anxiety inventory; CSS—cyberchondria severity scale; MSPSS—the multidimensional scale of perceived social support; QOLS—the quality of life scale.

**Table 2 ijerph-19-12962-t002:** Bivariate correlations.

	General (*n* = 538)					Females (*n* = 438)					Males (*n* = 100)				
	1.	2.	3.	4.	5.	1.	2.	3.	4.	5.	1.	2.	3.	4.	5.
1.Age	1					1					1				
2.SHAI	−0.048	1				−0.061	1				−0.072	1			
3.MSPSS	−0.015	−0.333 **	1			−0.029	−0.331 **	1			0.156	−0.190	1		
4.CSS	0.066	0.574 **	−0.141 **	1		0.031	0.627 **	−0.113 *	1		0.010	0.662 **	−0.024	1	
5.QOLS	0.161 **	−0.439 **	0.501 **	−0.199 **	1	0.154 **	−0.453 **	0.523 **	−0.170 **	1	0.254 *	−0.331 **	0.480 **	−0.282 **	1

* *p* < 0.05, ** *p* < 0.01; abbreviations: SHAI—the short health anxiety inventory; CSS—cyberchondria severity scale; MSPSS—the multidimensional scale of perceived social support; QOLS—the quality of life scale.

**Table 3 ijerph-19-12962-t003:** Bivariate correlations regarding education.

	Higher Education (*n* = 422)					Other Education (*n* = 422)				
	1.	2.	3.	4.	5.	1.	2.	3.	4.	5.
1.Age	1					1				
2.SHAI	−0.053	1				−0.041	1			
3.MSPSS	0.038	−0.263 **	1			−0.142	−0.363 **	1		
4.CSS	0.041	0.612 **	−0.084	1		0.003	0.743 **	−0.191 *	1	
5.QOLS	0.163 **	−0.382 **	0.465 **	−0.153 **	1	0.128	0.560 **	0.542 **	−0.415 **	1

* *p* < 0.05, ** *p* < 0.01; abbreviations: SHAI—the short health anxiety inventory; CSS—cyberchondria severity scale; MSPSS—the multidimensional scale of perceived social support; QOLS—the quality of life scale.

**Table 4 ijerph-19-12962-t004:** Results of mediation analysis.

	β	SE	95% CI	
			LLCI	ULCI
Direct effect of SHAI on CSS (a_1_)	0.639 *	0.034	0.573	0.705
Direct effect of SHAI on MSPSS (a_2_)	−0.298 *	0.042	−0.380	−0.217
Direct effect of CSS on QoL (b_1_)	0.056	0.045	−0.032	0.144
Direct effect of MSPSS on QoL (b_2_)	0.410 *	0.036	0.338	0.481
Direct effect of SHAI on QoL (c)	−0.328 *	0.047	−0.421	−0.236
Indirect effect (through CSS) of SHAI on QoL (a_1_b_1_)	0.036	0.030	−0.025	0.095
Indirect effect (through MSPSS) of SHAI on QoL (a_2_b_2_)	−0.122 *	0.025	−0.175	−0.078
Total indirect effect of SHAI on QoL (a_1_b_1_ + a_2_b_2_)	0.086 *	0.041	−0.167	−0.009

Significant effects (95%CI does not include zero) were marked with asterisks; covariates: age, gender and education; abbreviations: SHAI—the short health anxiety inventory; CSS—cyberchondria severity scale; MSPSS—the multidimensional scale of perceived social support; QOLS—the quality of life scale.

## Data Availability

The data analyzed during this study are included in this published article. Further inquiries can be directed to the corresponding authors.

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
