# Peer review of "Social Support Mediates the Association between Health Anxiety and Quality of Life: Findings from a Cross-Sectional Study"

_ijerph, 2022, doi:10.3390/ijerph191912962_

Round 1
Reviewer 1 Report
The manuscript entitled "Social support mediates the association between health anxiety and quality of life: Findings from a cross-sectional study" examines the mediating effects of social support and cyberchondria on the relationship between health anxiety and quality of life (QoL). Although the paper could have merit, there are several issues that must be resolved before this work is reconsidered for publication.
1. Introduction poorly describes variables in the mediation model. Unfortunately, nothing is known about direct associations (with direction and strength) between each pair of variables: health anxiety-cyberchondria; cyberchondria-QoL; health anxiety-social support; social support-QoL. In particular, the predictive value of variables used in the current study as predictors should be described based on previous literature. The current model of mediation is not justified at all. Not only should the predictions be reported from previous studies, but also a health-related theoretical model as a base for the mediation analysis. Some correlations between variables are not sufficient to assume the mediation model, which requires direct and causal effects of one variable on the other. Therefore, a solid theory is necessary to explain the mechanisms of interplay between variables. It must also be clearly stated, what is already known, what novelty the study offers, and how it can fill the scientific literature gap or mental health practice.
2. The authors did not mention the effects of age, education, and gender on anxiety, quality of life, social support, and cyberchondria. These associations must be shown with appropriate references, to understand their role as covariates in the mediation model. As a sensitivity analysis, gender differences could be performed for all continuous variables in the study.
3. It is necessary to show the situation of COVID-19 and restriction levels in Poland during the data collected. It is surprising that participants were recruited for seven months during the first and second pandemic waves. The pandemic situation dynamically changed during that long period. This is a serious flaw in data collection, which must be discussed and considered a limitation in the study.
4. More information should be added to the description of the QOLS. First of all, the information about Polish adaptation is missing. Please add the references or describe how the Polish translation and adaptation were performed. Moreover, Cronbach's alpha must be reported for the previous studies and the current sample.
5. Descriptive statistics are insufficiently reported. Apart from means and standard deviations, the other statistics must be reported and commented on for each variable in the mediation model, including a range of scores, median, skewness, kurtosis, and tests examining normal assumptions (e.g., Shapiro-Wilk or Kolmogorov-Smirnov). The other assumptions (than normal distribution) for multiple linear regression analysis (as a type of mediation analysis) should also be reported, including multicollinearity (Tolerance, VIF), homoscedasticity, multivariate normality, and autocorrelation.
6. Figure 1 should be presented at the end of the Introduction section, with hypotheses about the direct and indirect associations between variables. In the section on Results, the mediation model should be extended to standardized regression coefficients (betas), with p-values marked by asterisks.
7. All tables in the manuscript are unclear and require rearrangement. Please see the APA style guideline to learn what statistics should be reported and how to do it to maximize transparency and meet scientific standards.
8. The discussion should be extended to new information from the introduction and results.
Author Response
Thank you for taking the time and effort necessary to provide such insightful guidance. We carefully considered all the comments and hope that the revisions will meet your expectations and improve the quality of the paper.
Please find the response attached (as a pdf file).

Reviewer 2 Report
Summary
The current study used a cross-sectional, self-reported survey design and a parallel mediation model to assess perceived social support and cyberchondria as mediators in the relationship between health anxiety and quality of life. The article provides sufficient review of previous evidence for the importance of exploring the mechanisms underlying the relationship between health anxiety and quality of life. The constructs were explained well, creating a smooth transition from prior research to the current aims and hypotheses. The statistical model and the findings were clearly presented. Another strength is that the sample was large, allowing for rich data to be reported. Authors concluded that health anxiety demonstrated significant direct effects on quality of life, cyberchondria, and perceived social support. Further, the authors found that health anxiety indirectly affects quality of life through perceived social support. However, cyberchondria did not emerge as a significant mediator between health anxiety and quality of life. This study contributes to the literature by expanding on how health anxiety and cyberchondria are similar but differentially impact an individual’s quality of life. The findings can inform clinical interventions related to health anxiety and emphasizes the importance of how perceived social support may alleviate significantly distressing internet searching for individuals who ruminate about their health status.
Introduction
· Elaborating on the temporal precedence of health anxiety and cyberchondria may provide a clearer rationale for the placement of the variables within the model.
o There is evidence defining cyberchondria as a construct that involves the combination of health anxiety and compulsive internet searching. Given this, it seems logical to say and should be stated that health anxiety is an underlying component of cyberchondria and that these constructs would be highly correlated
o Evidence that highlights the bidirectionality of the relationship between compulsive internet searching and health anxiety should be reported. Understanding whether health anxiety precedes compulsive internet searching or vice versa would provide a clearer rationale as to why these constructs should be examined separated.
o Once this is elucidated, it would be helpful to hear the authors’ rationale for selecting health anxiety as a predictor and cyberchondria as a moderator.
Methods
· The current study used a snowball sampling procedure for participant recruitment. Please provide the rationale for selecting this recruitment method other than that it was “applied to involve adult representatives of the Polish population of internet users” (lines 88 -89).
o Snowball sampling may limit the representation of individuals who experience cyberchondria or health anxiety. If previous participants are tasked with recruiting others, then they may select individuals who they know engage in compulsive internet searching rather than just internet users. The lack of random selection poses threats to external validity which should be identified a limitation.
o The authors also mention that questionnaires were administered “using social backgrounds of the researchers” (lines 91-92). Please provide clarification on this administration method and the rationale.
· The authors report that they received approval from the Ethics Committee at Wroclaw Medical University in Poland (lines 96-97). Please clarify if this approval process is equivalent to an Institutional Review Board (IRB) approval, with further information regarding informed consent, participant risks/benefits, absence of coercion, etc.
· Authors collected data between May and December 2020. It is possible that data from the earlier recruitment may indicate higher scores on health anxiety and cyberchondria measures as well as lower scores on perceived social support than later data due to the governmental regulations occurring during this time (e.g., social distancing, limiting the amount of time spent outside the home, less publicly available information on implications of COVID and vaccines, etc.). It would be important to add a statement acknowledging this in the discussion to frame the findings and discuss the potential limitation of the data collection period on the data interpretation.
· The only measure that cued participants to consider a time frame for item rating was on the SHAI. The time frame for other measures should be listed as well to further specify the period during which participant experienced the measured construct. This relates to the previously stated limitation regarding data collection at the start of the pandemic (e.g., if participants recruited in May 2020 were cued to consider the items based on the preceding 6 months, how would this influence the results considering that the pandemic was not internationally declared until March 2020?).
· The authors did not list the psychometrics for the Polish version of the QOLS, while they did for all other measures. Psychometrics for all measures utilized in the study should be reported for consistency.
· On the SHAI, the authors reported that a score of 20 represented an optimal cutoff score. More information would be helpful to clarify the interpretation of the cutoff score (e.g., is a score of 20 points equivalent to high health anxiety? How would this compare to a score of 15? What scores would indicate a normative level of health anxiety as opposed to a clinically significant or elevated range?).
· Figure 1 provided a useful depiction of the parallel mediation model, but it did not include the coefficients for each path that was explored. These coefficients are commonly depicted and provide a more efficient way for readers to interpret the results. Additionally, the perceived social support variable in Figure 1 labels “social support” and should be changed to “perceived social support” considering the differentiation of these constructs cited in the introduction (lines 48-55), to avoid contradiction.
Results
· Age, gender, and education were included in the analysis as covariates without sufficient rationale or literature support.
o More information about these demographic variables in relation to the primary variables of interest (e.g. cyberchondria, health anxiety etc.) should be provided to explain why they were chosen as covariates.
o Also, while the authors included that older age was associated with greater quality of life (lines 162-164), no information regarding the relationship between the primary examined variables and demographic covariates were reported. Even if all the other correlations were not significant this should be stated.
Discussion
· The authors state that cyberchondria and health anxiety differ in that “cyberchondria causes distinctive deterioration in occupational, social, and family functioning together with misusing medical services when compared to health anxiety,” (lines 216-219). They then add that this distinction “…suggest[s] that potential compromised life quality among individuals with cyberchondria may be somehow buffered,” (lines 219-220), offering this as an explanation for why cyberchondria was not a significant moderator.
o Please explain how low quality of life is buffered by cyberchondria, considering that it results in several areas of functional impairment. If these impairments result from cyberchondria, it would seem that these individuals would have lower quality of life when compared with people who just experience health anxiety.
· The authors link this interpretation to cognitive behavioral theory (line 223) in that compulsive internet searching serves as a maladaptive coping mechanism that models negative reinforcement because this reassurance seeking behavior provides the individual with relief. However, this explanation contradicts cognitive behavioral theory because maladaptive coping mechanisms engender impairments in functioning, and then subsequently may reduce quality of life. Please explain the utility of this theory in interpreting the findings.
· The authors report how the results of this study do not provide information in terms of the COVID-19 pandemic, and instead frame it towards global crises in general. However, the specific occurrence of the COVID-19 pandemic appeared to be a catalyst for this type of research and informed their decision to include previous research that examined COVID-19 related constructs (lines 55-77). Furthermore, the data collection occurred during this specific pandemic. The discussion section addresses how the results of this study may be conceptually interpreted in the context of the current pandemic to provide practical information for clinicians and researchers in this area.
Author Response

(The authors gave the same response as above.)

Round 2
Reviewer 1 Report
The Authors answered all questions and improved the manuscript as suggested.
Reviewer 2 Report
Thank you for taking the time to consider our comments and suggestions. I found your responses to be appreciative and non-defensive, and it's clear you spent time addressing each of our points. The extensive edits made improved the clarity and the quality of the paper. I would also like to note that I appreciated the thoughtful responses to explain your position as well as present data and/or published literature to support the answers. I do not have any further concerns or recommendations.